# A Fair Power Allocation Approach to OFDM-Based NOMA with Consideration of Clipping

**Tao Tang [1], Yulong Mao [2] and Guangmin Hu [3],***

[1] School of Information and Communication Engineering, University of Electronic Science and Technology of China, Chengdu 611731, China; terotongcn@gmail.com
[2] National Key Laboratory of Science and Technology on Communications, University of Electronic Science and Technology of China, Chengdu 611731, China; maoyulong5566@163.com
[3] School of Resources and Environment, University of Electronic Science and Technology of China, Chengdu 611731, China
* Correspondence: hgm@uestc.edu.cn; Tel.: +86-28-6183-0209

**Abstract:** Orthogonal frequency division multiplexing-based non-orthogonal multiple access (OFDM-NOMA) is a competitive solution to achieve a capacity gain over orthogonal frequency division multiplexing-based orthogonal multiple access (OFDM-OMA). However, a major drawback of OFDM-based systems is the high peak-to-average power ratio (PAPR). Clipping is widely used for PAPR reduction, but it will degrade the capacity performance. Motivated by this fact, a fair user signal power allocation approach to OFDM-NOMA with clipping is proposed, where the power allocation factor is selected from a fair region. This approach fulfills the demand that OFDM-NOMA capacity can always outperform OFDM-OMA capacity for each paired user, regardless of the user pairing criteria, making it applicable for implementation using any scheduling paradigms. Therefore, the proposed approach can also be viewed as a solution to address fairness for the cell edge user in OFDM-NOMA systems. Both the theoretical and numerical results indicate that, although the capacity performance of OFDM-NOMA and OFDM-OMA is decreased and restrained at a high signal-to-noise ratio (SNR) by clipping, applying the proposed approach on OFDM-NOMA can still meet the aforementioned demand. Besides, it is shown that both the lower and upper bounds of the fair region are increased with a decreasing of clipping ratio.

**Keywords:** OFDM-NOMA; clipping; capacity; fair power allocation region

## 1. Introduction

In the ongoing fifth-generation (5G) wireless networks, there are many challenging requirements such as high spectral efficiency, massive connectivity and resilience to frequency selectivity to fulfill the demands of ever-increasing traffic volumes, massive growth in connected devices and the broad diversity of use cases. Non-orthogonal multiple access (NOMA) is one of the promising technologies have emerged to address the challenges in 5G and beyond wireless communications systems [1], which is highly expected to increase system throughput and accommodate massive connectivity. Different from conventional orthogonal multiple access (OMA) schemes where one resource block was allotted to only one user, the multiple users in NOMA can be assigned to the same frequency, time, and code, but in different power domain to improve the spectrum efficiency [2]. NOMA enables simultaneous transmission of multiple users on the same degrees of freedom via superposition coding with different power levels. It has been shown that NOMA is able to support massive connections, to reduce communication latency, and to increase system spectral efficiency [3–5].

Despite these attractive advantages, NOMA has some challenging problems to be solved, such as advanced transmitter design and the trade-off between performance and receiver complexity [6].

Orthogonal frequency division multiplexing (OFDM) has been widely adopted in fourth-generation (4G) wireless communication due to its advantages in improving spectrum efficiency and combatting multipath effects. Thus, the combination of OFDM and NOMA seems an attractive and competitive solution for 5G [7] and several studies have been made.

### 1.1. Related Works

Recently, more and more researches on OFDM-based NOMA (OFDM-NOMA) have been done from different perspectives such as sum-rate maximization, user fairness, and transmitting power minimization. For example, an optimization problem of minimizing the total transmitting power of OFDM-NOMA is investigated under the quality of service requirements in [8]. A power allocation algorithm is proposed in [9], which is obtained from the solution of a non-convex optimization problem for the maximization of the weighted system throughput for multi-carrier NOMA (MC-NOMA) system. An optimal power allocation scheme for sum throughput maximization of NOMA system is proposed in [10]. When it comes to addressing fairness in OFDM-NOMA systems, a joint solution is provided in [11] to solve the sub-channel assignment and power allocation problems iteratively by using a matching algorithm, aiming at maximizing the weighted total sum rate while taking into account user fairness. The fairness in uplink OFDM-NOMA systems is evaluated using Jain's fairness index in [12], but it is not directly addressed in the problem formulation. However, to maximize the geometric mean of the long-term averaged user rates, the approaches proposed in [9–11] rely on appropriate user pairing and power allocation. Thus, not all users will have equal opportunity to be scheduled, and the capacities of the scheduled users can not always outperform which they can achieve in OMA. More recently, a power allocation approach named Fair-NOMA is proposed to address fairness for NOMA systems in [13] and further discussed in [14]. Fair-NOMA demonstrates that NOMA capacity can fundamentally always outperform OMA capacity for each user, as long as the power allocation factor values in a fair region. The fair power allocation region given in Fair-NOMA is derived based on the independent identically distributed (i.i.d.) channels of users. Therefore, all users will have an equal opportunity to be scheduled, which is completely "fair" from a time-sharing perspective. It is worth noting that, the power allocation region given in the Fair-NOMA scheme is not suitable for OFDM-NOMA systems with consideration of PAPR reduction, since it considers the case with single carrier only.

OFDM-NOMA is shown to be able to outperform OFDM-based OMA (OFDM-OMA) in the perspectives of spectrum efficiency, power efficiency, and fairness. However, the high peak-to-average power ratio (PAPR) of OFDM-based systems will drive power amplifiers at the transmitter into saturation, producing interference among the subcarriers that degrades the system performance and corrupts the spectrum of the signal [15]. To address this issue, many PAPR reduction techniques have been proposed, which can be broadly classified into three main categories: signal distortion techniques (e.g., clipping), multiple signaling and probabilistic techniques (e.g., partial transmission sequences, PTS), and coding techniques (e.g., Golay complementary sequences, GCS) [16]. Multiple signaling and probabilistic techniques as well as coding techniques may look elegant from the perspectives of bit error rate (BER) and power increase, however, most of them are impractical for OFDM-based systems due to high computational complexity. Therefore, for the sake of simplicity, clipping-based techniques are highly recommended in standards such as long term evolution (LTE) [17] and implemented by most commercial products to reduce the PAPR. Nevertheless, the unavoidable signal power degradation and distortion noise caused by clipping restrain the performance of OFDM-based system [18]. To the best of the authors' knowledge, the impact of clipping on the capacity of each paired user in OFDM-NOMA has not been studied.

### 1.2. Contributions

Taking into account clipping, this paper proposes a fair user signal power allocation approach to OFDM-NOMA systems, where the bounds of the fair power allocation region are derived as functions

of the clipping ratio. The "fair" here means selecting the power allocation factor from the derived region to fulfill the demand that OFDM-NOMA capacity can fundamentally always outperform OFDM-OMA capacity for each paired user. The unique features of our approach could be stated as follows. It considers the impact of clipping on the capacity performance of OFDM-NOMA, making it more practical for implementation. Moreover, the channel coefficients of users are viewed as i.i.d. random variables in our approach (i.e., the location of the user in the cell is not considered), thus all users have equal probability to be scheduled, regardless of user pairing criteria. Therefore, the proposed approach can be also viewed as a solution to address fairness for the cell edge user in OFDM-NOMA systems. The impacts of clipping on the capacity performance of OFDM-NOMA and the fair power allocation region are studied as well. The numerical results indicate that, although the capacities of the two users in OFDM-NOMA are decreased and restrained at a high signal-to-noise ratio (SNR) due to clipping, the fair power allocation approach can still fulfill the "fair" demand.

### 1.3. Organization

The rest of this paper is organized as follows. Section 2 introduces the system model. Section 3 derives the fair power allocation region for OFDM-NOMA with consideration of clipping. The impacts of clipping on the fair power allocation region are studied as well. Section 4 shows the numerical results and Section 5 concludes this paper.

### 2. System Model

A downlink OFDM-NOMA system with one base station (BS) and two paired users is considered. In practice, most of NOMA schemes assume that there are at most two users multiplexing via the same degrees of freedom (DOF) [5,19,20], which can reduce both the computational complexity and decoding delay at the receiver. Both the BS and the users are equipped with a single antenna. Let $S_i[k]$ denote the modulated symbol of the $i$th user ($U_i$) on the $k$th subcarrier, where $i \in \{1, 2\}$, $k \in \{0, ..., N-1\}$. The average energy of $S_i[k]$ is normalized to be $\mathbb{E}[|S_i[k]|^2] = 1$, where $\mathbb{E}(\cdot)$ denotes the expectation operation. Suppose that the channel condition of $U_1$ is better than $U_2$. The power allocation factor for $U_1$ is $\alpha$ and the total power of $S_1[k]$ and $S_2[k]$ is $P$. Thus, the superposition symbol is

$$S[k] = \sqrt{\alpha P} \cdot S_1[k] + \sqrt{(1-\alpha)P} \cdot S_2[k]. \tag{1}$$

After the $N$-point inverse discrete Fourier transformation (IDFT), the output $s(n)$ is clipped before transmission. Note that $S[k]$ is the linear superposition of the signals of the two users. Therefore, it is easy to infer that the distribution of PAPR of OFDM-NOMA systems is the same as conventional OFDM systems. Since the distribution of PAPR of OFDM-NOMA is not the focus of this paper, the approximate complementary cumulative distribution function (CCDF) expression in [21] is directly introduced as a model of PAPR estimation for OFDM-NOMA systems, and the empirical results are provided in Section 4 to validate this expression, which is

$$Prob(PAPR > \gamma) \approx 1 - exp\left\{-N\sqrt{\frac{\pi\gamma}{3}}e^{-\gamma}\right\}, \tag{2}$$

where $\gamma$ is the PAPR threshold.

For OFDM-based systems, the out-of-band radiation of the clipped power is generally caused when the clipping is performed on the oversampled OFDM signals. Then the bandpass filter is required to suppress the out-of-band radiation. The bandpass filtering leads to the significant PAPR regrowth. On the other hand, when the clipping is performed on the signals at Nyquist sampling rate, all the distortion noise falls in-band and the out-of-band radiation is avoided due to the deliberate clipping, assuming that the signal is linearly amplified [22]. Throughout this paper, we suppose that clipping is performed on the Nyquist-rate samples where all the distortion noise falls in-band, and the power amplifier operates in a linear region after clipping. Thus, the impact of out-of-band radiation may not

be considered. Note that we do not consider the cyclic prefix (CP) in the PAPR analysis since it does not affect the peak or the average power [23]. As a clipping model of the baseband signal, the clipped signal can be written as

$$\widetilde{s}_n \triangleq \begin{cases} s(n), & for \quad |s(n)| \leq A \\ Ae^{j\theta}, & for \quad |s(n)| > A \end{cases}, \tag{3}$$

where $A$ is the maximum permissible amplitude over which the signal is clipped, and $\theta$ is the phase of $s(n)$.

We consider a Rayleigh flat fading channel in this paper. The channel coefficients with respect to the two users are modeled as i.i.d. random variables with probability density function (PDF) $f_{|H|^2}(x) = \frac{1}{\rho}e^{-\frac{x}{\rho}}$, where $\rho = \mathbb{E}[|H|^2]$. Let $\widetilde{S}[k]$ represent the discrete Fourier transformation (DFT) of the clipped frequency-domain signals, then the received signal of each user is

$$Y_i[k] = \widetilde{S}[k]H_i[k] + Z_i[k], \ i = 1,2, \tag{4}$$

where $H_i[k]$ is the channel coefficient of the $k$th subcarrier corresponding to the $i$th user, and $Z_i[k]$ denotes the additive Gaussian white noise (AGWN) with zero mean and variance $\sigma_z^2 = 1$.

The successive interference cancellation (SIC) process is implemented at the receiver side, and the optimal order for detection is the ascending order of channel gain [1]. Without loss of generality, we assume that $|H_1[k]|^2 \geqslant |H_2[k]|^2$, the signals of $U_2$ are detected first by treating signals of $U_1$ as noise at the receiver of $U_2$. In other words, $U_2$ does not perform SIC. For the receiver of $U_1$, it has to detect the interference from the signals of $U_2$, and then the detected interference will be reconstructed and subtracted. After that, $U_1$ detects the signals of its own.

## 3. Fair Power Allocation Approach to OFDM-NOMA with Clipping

### 3.1. Fair Power Allocation Region of OFDM-NOMA with Clipping

Actually, the proposed fair power allocation approach is motivated by the demand that capacities of the two paired users in OFDM-NOMA with clipping can always outperform which they can achieve in OFDM-OMA with clipping, i.e.,

$$C_1^N(\alpha) \geq C_1^O, \ C_2^N(\alpha) \geq C_2^O, \tag{5}$$

where $\{C_1^N(\alpha), \ C_2^N(\alpha)\}$ and $\{C_1^O, \ C_2^O\}$ are the capacities of the two users in OFDM-NOMA and OFDM-OMA, respectively; and OFDM-OMA means that the BS takes up a half of the period of the same channel to transmit each user's signal [8].

The fair region of $\alpha$ that makes (5) true is defined as an interval, which is bounded by $\alpha_{\text{inf}}$ and $\alpha_{\text{sup}}$. To be specific, $\alpha_{\text{inf}}$ is the power allocation factor that gives $C_1^N(\alpha_{\text{inf}}) = C_1^O$, while $\alpha_{\text{sup}}$ is the one that gives $C_2^N(\alpha_{\text{sup}}) = C_2^O$. Note that $\alpha$ is a power allocation proportion with respect to $U_1$, it is easy to confirm that $C_1^N(\alpha)$ increases with the increase of $\alpha$, whereas $C_2^N(\alpha)$ decreases with the increase of $\alpha$ since the power allocation proportion to $U_2$ (i.e., $1 - \alpha$) is reduced. Hence, for any $\alpha \in [\alpha_{\text{inf}}, \alpha_{\text{sup}}]$, the inequalities in (5) always holds.

Before deriving the general expressions of $\alpha_{\text{inf}}$ and $\alpha_{\text{sup}}$, we would like to introduce the implementation of the fair power allocation approach first:

1. Acquire the channel state information (i.e., $H_i[k]$), the maximum permissible amplitude of the envelope limiter (i.e., $A$), and the transmitting power of superposition symbol (i.e., $P$);
2. Calculate the specific values of $\alpha_{\text{inf}}$ and $\alpha_{\text{sup}}$ by using the parameters in (1);
3. Select the power allocation factor from the region $[\alpha_{\text{inf}}, \alpha_{\text{sup}}]$ under a certain constraint (e.g., spectral efficiency constraint), then allocate power to the two paired users according to the selected $\alpha$.

Several selection algorithms for power allocation factor under a certain constraint have been proposed for OFDM-NOMA [3–6], and are shown to achieve a long-term averaged capacity gain over OFDM-OMA. However, those algorithms rely on appropriate user pairing, implying that cell edge users could not have equal opportunity to be scheduled. The main significance of our approach is to demonstrate that there always exists a fair power allocation region for any two users in the cell, making that OFDM-NOMA can fundamentally achieve a capacity gain over OFDM-OMA for each user. Thus, those selection algorithms can be adopted in our approach to address the fairness of cell edge users. Since the selection algorithm is not the focus of this paper, the rest of this subsection will concentrate on the derivation of the fair power allocation region.

According to the Bussgang theorem, the clipped signal $\tilde{s}(n)$ can be modeled as

$$\tilde{s}(n) = \varepsilon s(n) + d(n), \tag{6}$$

where $d(n)$ is i.i.d. random complex distortion with zero mean, but not Gaussian in general, and $\varepsilon$ is the linear attenuation factor given by [10]

$$\varepsilon = 1 - e^{-\beta^2} + \frac{\sqrt{\pi}\beta}{2} erfc(\beta), \tag{7}$$

where $\beta$ is the clipping ratio defined as $\beta \triangleq \frac{A}{\sqrt{\mathbb{E}[|s(n)|^2]}}$, and $erfc(\cdot)$ represents the complementary error function.

When the number of subcarrier $N$ is large, it follows from the central limit theorem that the time-domain signal $s(n)$ is approximately complex Gaussian distributed. Hence, the amplitude $|s(n)|$ is a Rayleigh random variable with probability density function given as

$$f_{|s(n)|}(x) = \frac{2x}{P} e^{-\frac{x^2}{P}}. \tag{8}$$

Combining (3), the average power of the clipped signal, $\widetilde{P}$, can be calculated as

$$\widetilde{P} = \mathbb{E}\left[|\tilde{s}_n|^2\right] = \int_0^\infty |\tilde{s}_n|^2 f_{|s(n)|}(x)dx = \left(1 - e^{-\beta^2}\right)P. \tag{9}$$

Referring to (6), the average power of $\varepsilon s(n)$ is

$$P_C = \varepsilon^2 \mathbb{E}\left[|s(n)|^2\right] = \varepsilon^2 P = \kappa\widetilde{P}, \tag{10}$$

where $\kappa$ is defined as $\kappa \triangleq P_C / \tilde{P}$. Combining (7) and (9), $\kappa$ can be expressed as a function of $\beta$, which is

$$\kappa = \frac{\left(1 - e^{-\beta^2} + \frac{\sqrt{\pi}\beta}{2} erfc(\beta)\right)^2}{1 - e^{-\beta^2}}. \tag{11}$$

Therefore, the average power of $d(n)$ in (6) is

$$P_D = \widetilde{P} - P_c = (1 - \kappa)\widetilde{P}. \tag{12}$$

We assume that $H_i[k]$ is statistically independent and quasi-statical during an OFDM symbol period, thus the index $k$ of $H_i[k]$ is dropped, i.e., $H_i = H_i[k]$. For $U_1$, the interference from $U_2$ is successfully eliminated due to perfect SIC. Therefore, only AGWN and distortion need to be considered for the calculation of the capacity of $U_1$, which is

$$
\begin{aligned}
C_1^N &= \log_2\left(1 + \frac{\alpha P_C |H_1|^2}{P_D |H_1|^2 + 1}\right) \\
&= \log_2\left(1 + \frac{\alpha\kappa\widetilde{P}|H_1|^2}{(1-\kappa)\widetilde{P}|H_1|^2 + 1}\right).
\end{aligned}
\tag{13}
$$

While for $U_2$, since no SIC is performed at the receiver, interference from $U_1$ exists. With the addition of AGWN and distortion, the capacity of $U_2$ is

$$C_2^N = \log_2\left(1 + \frac{(1-\alpha)P_C|H_2|^2}{\alpha P_C|H_2|^2 + P_D|H_2|^2 + 1}\right)$$
$$= \log_2\left(1 + \frac{(1-\alpha)\kappa\widetilde{P}|H_2|^2}{\alpha\kappa\widetilde{P}|H_2|^2 + (1-\kappa)\widetilde{P}|H_2|^2 + 1}\right). \tag{14}$$

According to (5), the capacities of the two users in OFDM-OMA should be

$$C_i^O = \frac{1}{2}\log_2\left(1 + \frac{P_C|H_i|^2}{P_D|H_i|^2 + 1}\right)$$
$$= \frac{1}{2}\log_2\left(1 + \frac{\kappa\widetilde{P}|H_i|^2}{(1-\kappa)\widetilde{P}|H_i|^2 + 1}\right), \tag{15}$$

where $i = 1, 2$.

With the combination of (5), (13), (14) and (15), the bounds of the fair power allocation region are obtained, which are

$$\alpha_{\text{inf}} = \frac{\left(\left(\frac{\widetilde{P}|H_1|^2+1}{(1-\kappa)\widetilde{P}|H_1|^2+1}\right)^{\frac{1}{2}} - 1\right)\left((1-\kappa)\widetilde{P}|H_1|^2 + 1\right)}{\kappa\widetilde{P}|H_1|^2}, \tag{16}$$

$$\alpha_{\text{sup}} = \frac{\left(\left(\frac{\widetilde{P}|H_2|^2+1}{(1-\kappa)\widetilde{P}|H_2|^2+1}\right)^{\frac{1}{2}} - 1\right)\left((1-\kappa)\widetilde{P}|H_2|^2 + 1\right)}{\kappa\widetilde{P}|H_2|^2}. \tag{17}$$

### 3.2. Effects of Clipping on The Capacity Performance of OFDM-NOMA

From (13) and (14), it is easy to verify that the capacities of $U_1$ and $U_2$ are reduced due to the distortion and power reduction caused by clipping. Furthermore, we are also interested in the asymptotic capacity performance of each paired user in OFDM-NOMA when the signal-to-noise ratio (SNR) towards infinity (i.e., $P \to \infty$). Since $\tilde{P} \propto P$, the asymptotic expressions could be easily extracted from (13) and (14) by following the L'Hospital's rule, which are

$$\lim_{\widetilde{P}\to\infty} C_1^N = \lim_{\widetilde{P}\to\infty} \log_2\left(1 + \frac{\alpha\kappa\widetilde{P}|H_1|^2}{(1-\kappa)\widetilde{P}|H_1|^2 + 1}\right)$$
$$= \log_2\left(\frac{1 - (1-\alpha)\kappa}{1 - \kappa}\right), \tag{18}$$

$$\lim_{\widetilde{P}\to\infty} C_2^N = \lim_{\widetilde{P}\to\infty} \log_2\left(1 + \frac{(1-\alpha)\kappa\widetilde{P}|H_2|^2}{\alpha\kappa\widetilde{P}|H_2|^2 + (1-\kappa)\widetilde{P}|H_2|^2 + 1}\right)$$
$$= \log_2\left(\frac{1}{1 - (1-\alpha)\kappa}\right). \tag{19}$$

According to (18) and (19), given $\alpha$ and $\kappa$ ($\kappa \neq 1$), the asymptotic capacities of $U_1$ and $U_2$ are fixed values. In other words, the capacity of OFDM-NOMA for each user is not only reduced but also restrained at high SNR region by clipping. As a result, improving the transmitting power may not be an effective way to achieve a capacity gain for OFDM-NOMA with clipping.

### 3.3. Effects of Clipping on The Fair Power Allocation Region of OFDM-NOMA

When using the hard envelope limiter defined in [24], i.e., $\kappa_0 = \lim_{\beta \to 0} \kappa = \frac{\pi}{4}$, $C_1^N$ and $C_2^N$ have the worse performance degradation; when no clipping is needed, and the power amplifier operates in a linear region, $\kappa_\infty = \lim_{\beta \to \infty} \kappa = 1$. Thus, the range of $\kappa$ is $[\pi/4, 1]$. As functions of $\kappa$, both $\alpha_{\inf}(\kappa)$ and $\alpha_{\sup}(\kappa)$ have the same form except the channel coefficients. In other words, $\alpha_{\inf}(\kappa)$ has the same monotonicity as $\alpha_{\sup}(\kappa)$, which is demonstrated in Theorem 1.

**Theorem 1.** *For OFDM-NOMA using clipping, given the total power allocated to each subcarrier (i.e., P) and the channel coefficients of the two paired users (i.e., $H_1$ and $H_2$), $\forall \kappa \in [\pi/4, 1]$, $\alpha'_{inf}(\kappa) < 0$ and $\alpha'_{sup}(\kappa) < 0$.*

**Proof.** Since $\alpha_{\inf}(\kappa)$ has the same form as $\alpha_{\sup}(\kappa)$, which is

$$f(x) = \frac{\left( \left( \frac{B+1}{(1-x)B+1} \right)^{\frac{1}{2}} - 1 \right) ((1-x)B+1)}{Bx}, \tag{20}$$

where $B$ represents $\tilde{P}|H_1|^2$ or $\tilde{P}|H_2|^2$. Let $f(x) = f_1(x) f_2(x)$, where

$$f_1(x) = \left( \frac{B+1}{(1-x)B+1} \right)^{\frac{1}{2}} - 1, \tag{21}$$

and

$$f_2(x) = \frac{(1-x)B+1}{Bx}. \tag{22}$$

Then we have

$$f_1'(x) = -\frac{B}{2}(B+1)^{\frac{1}{2}} ((1-x)B+1)^{-\frac{3}{2}}, \tag{23}$$

and

$$f_2'(x) = -\frac{B^2+B}{(Bx)^2}. \tag{24}$$

Clearly, for all $x \in [\pi/4, 1]$, $1 - x \geqslant 0$, and note that $B > 0$, it is easy to confirm that

$$f_1'(x) < 0, \; f_2'(x) < 0. \tag{25}$$

Since $f_1(x) > 0$, $f_2(x) > 0$, then $f'(x) = f_1'(x) f_2(x) + f_1(x) f_2'(x) < 0$. Hence, for all $\kappa \in [\pi/4, 1]$, $\alpha'_{\inf}(\kappa) < 0$, $\alpha'_{\sup}(\kappa) < 0$. $\square$

According to Theorem 1, the lower bound $\alpha_{\inf}$ and the upper bound $\alpha_{\sup}$ are monotone decreasing functions of $\kappa$. Especially, if no clipping is needed, and the amplifier operates in a linear region, i.e., $\kappa = 1$, $\alpha_{\inf}$ reaches its minimum $\alpha_{\inf}^{min}$; while clipping is performed, i.e., $\pi/4 \leqslant \kappa < 1$, $\alpha_{\inf} > \alpha_{\inf}^{min}$. With this result in mind, and recall that the lower bound makes $C_1^N(\alpha_{\inf}) = C_1^O$, one can conclude that clipping leads to more power consumption compared with the case without clipping, to guarantee that OFDM-NOMA can achieve a rate as good as OFDM-OMA for $U_1$.

While for $U_2$, the fair power allocation region should be $[1 - \alpha_{\sup}, 1 - \alpha_{\inf}]$, and note that $C_2^N(\alpha_{\sup}) = C_2^O$ holds when the power allocation proportion for $U_2$ equals to $1 - \alpha_{\sup}$. Similarly, if no clipping is needed, $\alpha_{\sup}$ reaches its minimum value $\alpha_{\sup}^{min}$ as well, such that $1 - \alpha_{\sup}$ reaches the maximum. Therefore, on the contrary, to guarantee that OFDM-NOMA can achieve a rate as good as OFDM-OMA for $U_2$, the case with clipping requires less power than which with no need of clipping.

For purpose of performance evaluation, the ergodic capacities of $U_1$ and $U_2$ in OFDM-NOMA and OFDM-OMA are derived in this section. Since the ergodic capacities of the two users in OFDM-NOMA

depend on the power allocation factor $\alpha$, the lower bound of the ergodic capacity is derived and compared with which in OFDM-OMA for each user, to evaluate the performance of the proposed fair power allocation approach. Note that the channels of the two users are i.i.d. random variables, thus the joint PDF of the channel gains of the two users is

$$f_{|H_1|^2,|H_2|^2}(x_1, x_2) = \frac{2}{\rho^2} e^{-\frac{x_1+x_2}{\rho}}. \tag{26}$$

Then the ergodic capacity of each user in OFDM-OMA is

$$
\begin{aligned}
\mathbb{E}[C_1^O] &= \int_0^\infty \int_{x_2}^\infty \frac{1}{\rho^2} e^{-\frac{x_1+x_2}{\rho}} \cdot \log_2\left(1 + \frac{\kappa \tilde{P} x_1}{(1-\kappa)\tilde{P} x_1 + 1}\right) dx_1 dx_2 \\
&= \int_0^\infty \int_{x_2}^\infty \frac{1}{\rho^2} e^{-\frac{x_1+x_2}{\rho}} \cdot \log_2\left(\frac{\tilde{P} x_1 + 1}{(1-\kappa)\tilde{P} x_1 + 1}\right) dx_1 dx_2,
\end{aligned}
\tag{27}
$$

$$
\begin{aligned}
\mathbb{E}[C_2^O] &= \int_0^\infty \int_0^{x_1} \frac{1}{\rho^2} e^{-\frac{x_1+x_2}{\rho}} \cdot \log_2\left(1 + \frac{\kappa \tilde{P} x_2}{(1-\kappa)\tilde{P} x_2 + 1}\right) dx_2 dx_1 \\
&= \int_0^\infty \int_0^{x_1} \frac{1}{\rho^2} e^{-\frac{x_1+x_2}{\rho}} \cdot \log_2\left(\frac{\tilde{P} x_2 + 1}{(1-\kappa)\tilde{P} x_2 + 1}\right) dx_2 dx_1.
\end{aligned}
\tag{28}
$$

In the case of OFDM-NOMA using the proposed fair power allocation approach with $\alpha = \alpha_{\inf}$, where $C_1^N(\alpha_{\inf}) = C_1^O$, it is easy to infer that $\mathbb{E}\left[C_1^N(\alpha_{\inf})\right]$ equals to $\mathbb{E}\left[C_1^O\right]$. Similarly, the lower bound of the ergodic capacity of OFDM-NOMA for $U_2$ is reached when $\alpha = \alpha_{\sup}$, i.e., $\mathbb{E}\left[C_2^N(\alpha_{\sup})\right] = \mathbb{E}\left[C_2^O\right]$. The following derivation also validates our deduction.

$$
\begin{aligned}
\mathbb{E}[C_1^N(\alpha_{\inf})] &= \int_0^\infty \int_{x_2}^\infty \frac{1}{\rho^2} e^{-\frac{x_1+x_2}{\rho}} \cdot \log_2\left(1 + \frac{\alpha_{\inf}\kappa \tilde{P} x_1}{(1-\kappa)\tilde{P} x_1 + 1}\right) dx_1 dx_2 \\
&= \int_0^\infty \int_{x_2}^\infty \frac{1}{\rho^2} e^{-\frac{x_1+x_2}{\rho}} \cdot \log_2\left(\frac{\tilde{P} x_1 + 1}{(1-\kappa)\tilde{P} x_1 + 1}\right) dx_1 dx_2,
\end{aligned}
\tag{29}
$$

$$
\begin{aligned}
\mathbb{E}[C_2^N(\alpha_{\sup})] &= \int_0^\infty \int_0^{x_1} \frac{1}{\rho^2} e^{-\frac{x_1+x_2}{\rho}} \cdot \log_2\left(1 + \frac{(1-\alpha_{\sup})\kappa \tilde{P} x_2}{\alpha_{\sup}\kappa \tilde{P} x_2 + (1-\kappa)\tilde{P} x_2 + 1}\right) dx_2 dx_1 \\
&= \int_0^\infty \int_0^{x_1} \frac{1}{\rho^2} e^{-\frac{x_1+x_2}{\rho}} \cdot \log_2\left(\frac{\tilde{P} x_2 + 1}{(1-\kappa)\tilde{P} x_2 + 1}\right) dx_2 dx_1.
\end{aligned}
\tag{30}
$$

Although the ergodic capacity expressions of the two paired users are not in closed form, they can be calculated numerically.

## 4. Numerical Results

### 4.1. Comparison of CCDFs of PAPR in Conventional OFDM and OFDM-NOMA

As shown in Figure 1, both the approximate CCDF for conventional OFDM systems and empirical CCDF of PAPR in OFDM-NOMA systems with 128 and 1024 subcarriers per OFDM symbol block are given. The horizontal and vertical coordinates represent the threshold for the PAPR and the probability of the PAPR of a data block exceeding the threshold, respectively. Each paired user's signals are modulated with 4-QAM. The inference in Section 2 that OFDM-NOMA has the same distribution of PAPR as conventional OFDM is validated.

### 4.2. Lower and Upper Bounds of Fair Power Allocation Region for OFDM-NOMA with Clipping

Without loss of generality, the signal-to-noise ratio (SNR) and the channel gain ratio are set to be 10 dB. As shown in Figure 2, both the lower and upper bounds of the fair power allocation region for $U_1$ decrease with an increase of $\kappa$, which matches Theorem 1. More specifically, given the total transmitting power $P$, the presence of clipping leads to a greater proportion of power allocating to

$U_1$ compared to the case of no clipping, to fulfill the demand that $U_1$ in OFDM-NOMA can achieve a rate as good as which it can achieve in OFDM-OMA. On the contrary, to fulfill such demand for $U_2$, the power allocation proportion of $U_2$ (i.e., $1 - \alpha_{\sup}$) in OFDM-NOMA with clipping is less than that in OFDM-NOMA with no need of clipping, e.g, $\alpha_{\sup}(\kappa = 0.85) \approx 0.45$. Thus the power allocation proportion of $U_2$ should be $1 - \alpha_{\sup}(\kappa = 0.9) = 0.55$, which is 0.6 in the case of no clipping.

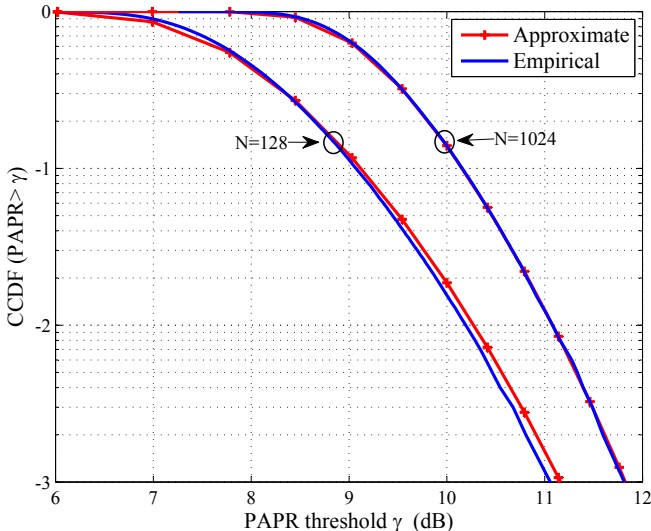

**Figure 1.** The approximate CCDF and empirical CCDF of PAPR of an OFDM-NOMA system with 128 and 1024 subcarriers.

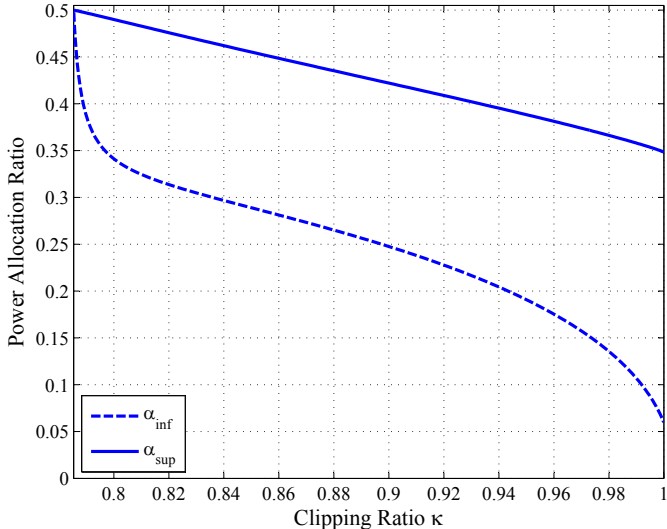

**Figure 2.** Lower and upper bounds of power allocation region for Fair OFDM-NOMA versus clipping ratio.

### 4.3. Comparison of Ergodic Capacity of OFDM-NOMA and OFDM-OMA

In Figure 3, ergodic capacities of $U_1$ and $U_2$ in OFDM-NOMA and OFDM-OMA with clipping (i.e., $\kappa = 0.8$) are given as well as the case without clipping (i.e., $\kappa = 1$). The horizontal axis represents the input SNR, where $SNR = P/\sigma_z^2$. The ergodic capacity of $U_1$ in OFDM-NOMA is given with power allocation factor $\alpha = \alpha_{\inf}$ whereas that of $U_2$ is given with $\alpha = \alpha_{\sup}$, aiming to verify the performance of the fair power allocation approach.

On one hand, the ergodic capacities of $U_1$ and $U_2$ are reduced by clipping due to the distortion and power degradation. Especially at high SNR region, the ergodic capacities of the two users are limited to about 1.2 bits/s/Hz. That is to say, it is not theoretically possible to achieve error-free performance

with the information data rate of 1.2 bits/s/Hz for $U_1$ and $U_2$. On the other hand, though clipping restrains the capacity performance, the fair power allocation approach to OFDM-NOMA is shown to be capable to ensure the two users achieving a rate at least as good as they can achieve in OFDM-OMA.

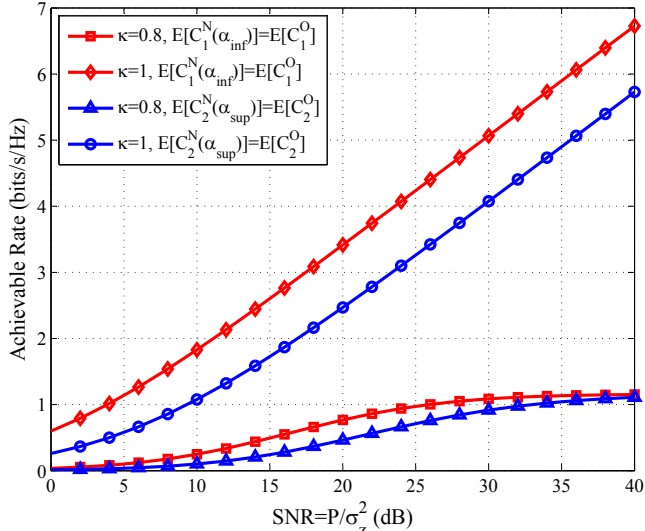

**Figure 3.** Comparing the capacity of OFDM-NOMA using fair power allocation approach vs. OFDM-OMA.

## 5. Conclusions

This paper proposes a fair user signal power allocation approach to OFDM-NOMA systems with clipping, to fulfill the demand that OFDM-NOMA capacity can fundamentally always outperform OFDM-OMA capacity for each paired user. It must be stressed that, all users in the cell can have equal opportunity to be scheduled by applying the proposed approach on OFDM-NOMA, regardless of the user pairing criteria and scheduling scheme. Thus, the fair power allocation approach is a recommended solution to address fairness for the cell edge users. Besides, the impacts of clipping on the capacity performance of OFDM-NOMA and the fair power allocation region are studied as well.

**Author Contributions:** Conceptualization, T.T. and Y.M.; methodology, T.T.; software, Y.M.; validation, Y.M.; formal analysis, T.T.; investigation, T.T.; resources, T.T.; data curation, Y.M.; writing–original draft preparation, T.T.; writing–review and editing, T.T.; visualization, T.T.; supervision, G.H.; project administration, G.H.; funding acquisition, T.T. All authors have read and agreed to the published version of the manuscript.

**Funding:** This research was funded by the National Natural Science Foundation of China grant number 0561701074.

**Conflicts of Interest:** The authors declare no conflict of interest.

## Abbreviations

The following abbreviations are used in this manuscript:

| | |
|---|---|
| OFDM-NOMA | Orthogonal frequency division multiplexing based non-orthogonal multiple access |
| OFDM-OMA | Orthogonal frequency division multiplexing based orthogonal multiple access |
| PAPR | Peak-to-average power ratio |
| SNR | Signal-to-noise ration |
| 5G | Fifth-generation |
| NOMA | Non-orthogonal multiple access |
| OMA | Orthogonal multiple access |
| OFDM | Orthogonal frequency division multiplexing |
| 4G | Fourth-generation |
| MC-NOMA | Multi-carrier non-orthogonal multiple access |

| PTS | Partial transmission sequences |
|-----|-------------------------------|
| GCS | Golay complementary sequences |
| BER | Bit error rate |
| LTE | Long term evolution |
| BS | Base station |
| DOF | Degrees of freedom |
| IFDT | Inverse discrete Fourier transformation |
| CCDF | Complementary cumulative distribution function |
| CP | Cyclic prefix |
| PDF | Probability density function |
| DFT | Discrete Fourier transformation |
| AGWN | Additive Gaussian white noise |
| SIC | Successive interference cancellation |

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
