# Peer review of "A Fair Power Allocation Approach to OFDM-Based NOMA with Consideration of Clipping"

_electronics, doi:10.3390/electronics9101743_

Round 1

Reviewer 1 Report

The paper is well-written and contains substantial results. Authors show that choosing suitable clipping parameters, non-orthogonal multiple access allows always achieve better throughput than the classical orthogonal multiple access.

There is good overview of the state of the art in orthogonal frequency division multiplexing based non-orthogonal multiple access (OFDM-NOMA) and orthogonal frequency division multiplexing based orthogonal multiple access (OFDM-OMA).

As novel solution, authors propose a fair user signal power allocation approach to OFDM-NOMA. Among other results they show that this approach always outperforms the classical OFDM-OMA.

The paper is well-structured, the references are adequate and it contains interesting results. So I recommend it for publication.

Reviewer 2 Report

A fair user signal power allocation approach to OFDM-NOMA with clipping has been proposed by the authors. The topic is timely and proposed approach is interesting. Reviewer has the following major comments. 1) The backgrounds of NOMA based networks are not clearly described. Authors can refers some latest literature for this such as a) "Resource Optimization in Full Duplex Non-Orthogonal Multiple Access Systems," in IEEE Transactions on Wireless Communications, vol. 18, no. 9, pp. 4312-4325, Sept. 2019 and "An Energy-Efficient Approach Towards Power Allocation in Non-Orthogonal Multiple Access Full-Duplex AF Relay Systems," 2018 IEEE 19th International Workshop on Signal Processing Advances in Wireless Communications (SPAWC), Kalamata, 2018. 2) Authors must discuss about complexity analysis similar to "QoS-Driven Resource Allocation and EE-Balancing for Multiuser Two-Way Amplify-and-Forward Relay Networks," in IEEE Transactions on Wireless Communications, vol. 16, no. 5, pp. 3189-3204, May 2017. 

Reviewer 3 Report

The article proposes a power allocation approach for OFDM-NOMA systems in the presence of clipping providing analytical derivations. A fundamental question is: how general is the approach, i.e., can it be scaled to more than a pair (two) of users, and what would be the performance?

In the Introduction, the authors could reference the ongoing 5G standardization, and therefore, position their work with respect to that. I would also add some more intuitive explanation for NOMA vs OMA for the less familiar reader.

Something does not sound correct in the phrase (page 2, line 57): 'Multiple signaling and probabilistic techniques as well as coding techniques may look elegant from the perspectives of bit error rate (BER) and power increase, however, most of them are not impractical for OFDM-based systems due to high computational complexity.' Do you mean 'are not practical'?

Could the authors better clarify, e.g. by examples or references, the following concept: 'when clipping is performed on the signals at the Nyquist sampling rate all the distortion noise falls in-band'?

Round 2

Reviewer 2 Report

No more comments.